# Organized Chaos: Deciphering Immune Cell Heterogeneity’s Role in Inflammation in the Heart

**DOI:** 10.3390/biom12010011

**Published:** 2021-12-22

**Authors:** Alexa Corker, Lily S. Neff, Philip Broughton, Amy D. Bradshaw, Kristine Y. DeLeon-Pennell

**Affiliations:** 1Department of Medicine, Division of Cardiology, Medical University of South Carolina, Charleston, SC 29425, USA; corker@musc.edu (A.C.); neffli@musc.edu (L.S.N.); broughph@musc.edu (P.B.); bradshad@musc.edu (A.D.B.); 2Ralph H. Johnson Veterans Affairs Medical Center, Medical University of South Carolina, Charleston, SC 29401, USA

**Keywords:** cardiovascular disease, myocardial infarction, pressure overload, inflammation

## Abstract

During homeostasis, immune cells perform daily housekeeping functions to maintain heart health by acting as sentinels for tissue damage and foreign particles. Resident immune cells compose 5% of the cellular population in healthy human ventricular tissue. In response to injury, there is an increase in inflammation within the heart due to the influx of immune cells. Some of the most common immune cells recruited to the heart are macrophages, dendritic cells, neutrophils, and T-cells. In this review, we will discuss what is known about cardiac immune cell heterogeneity during homeostasis, how these cell populations change in response to a pathology such as myocardial infarction or pressure overload, and what stimuli are regulating these processes. In addition, we will summarize technologies used to evaluate cell heterogeneity in models of cardiovascular disease.

## 1. Introduction

The healthy mammalian heart contains an estimated 2–3 billion cardiac myocytes, which is approximately 75% of normal myocardial tissue volume but only one third of total cell numbers [1,2]. Other cell types include fibroblasts, resident macrophages, endothelial cells, and perivascular cells [1,2]. Resident immune cells originate from progenitor cells during development and compose 5% of the cellular population in healthy human ventricular tissue [3,4]. During cardiac development, resident cardiac macrophages are thought to facilitate the conduction system and remove apoptotic cells [5]. In healthy mouse cardiac tissue, the number of mononuclear phagocytes, neutrophils, B-cells, and T-cells is 12-fold more than in the skeletal muscle demonstrating the importance of immune cells in maintaining heart homeostasis [6]. Adding to this complexity, Squiers et al. showed that there are sex-dependent differences in cell composition of mouse hearts; male mice have more myeloid cells, granulocytes, and B-cells while female mice have more NK cells and T-cells [7].

In response to injury after myocardial infarction (MI), there is an increase in inflammation within the heart due to the influx of immune cells. Some of the most common immune cells recruited to the heart are macrophages, dendritic cells, monocytes, neutrophils, and T-cells [1,5]. Inflammation after MI must reside within a Goldilocks zone to have a beneficial effect. The Goldilocks zone is defined as the balance of immune cells necessary for efficient cardiac healing. While some inflammation is essential for clearance of tissue damage, excessive or prolonged inflammation can be detrimental as it may lead to increased cardiac rupture from disproportionate collagen degradation, infarct expansion via phagocytosis of healthy cardiomyocytes, and/or increased dilation of the left ventricle (LV; Figure 1A) [5,8]. Knowing the stimuli that facilitates recruitment and phenotypic changes within cell types can help uncover regulators of beneficial versus adverse cardiac remodeling [9].

Heart failure with preserved ejection fraction (HFpEF) is a diagnosis frequently associated with conditions that give rise to increased pressure overload (PO) on the heart including hypertension and aortic valve disease. A hallmark of HFpEF is an increase in myocardial stiffness contributed by both changes in cardiac myocyte sarcomeres and increases in interstitial collagen content [10]. More recently, there has been a shift in the focus to include the role of immune cells in regulating HFpEF pathology. While the role of immune cells post-MI is better understood, fewer studies have focused on how these immune cells regulate each other in the heart before and after PO.

In this review, we will discuss what is known about cardiac immune cell heterogeneity during homeostasis and in response to injury and what is known regarding stimuli that regulate this process (Figure 1B). In addition, we will summarize technologies used to evaluate cell heterogeneity in models of cardiovascular disease.

## 2. Immune Cell Heterogeneity in the Heart

In order to identify new drug targets and possible biomarkers, we must first understand the differences in cellular composition between healthy and diseased states. The immune system plays a vital role in regulating cardiac health and homeostasis [5]. Cardiac remodeling in diseased hearts is temporal and spatially dependent with unresolved or mistimed inflammation leading to excessive matrix degradation and cardiac rupture, dilation of the LV, or development of heart failure [11,12]. Ensuring that the immune and fibrotic systems are balanced is key for effective cardiac healing after injury. Unraveling the roles of each immune cell type and their interactions with each other is important to understand the mechanisms of cardiac wound healing. This will allow for a better understanding of cellular alterations that occur in chronic inflammation and hypertrophic scarring so that effective therapeutic solutions for beneficial cardiac healing can be developed.

### 2.1. Macrophages

#### 2.1.1. Macrophages at Homeostasis

Macrophages are innate immune cells that can detect, phagocytose, destroy bacteria/viruses, and mediate injury. Resident cardiac macrophages make up a large proportion of the non-cardiomyocyte cell population compared to other immune cell types, such as lymphocytes [13]. Cardiac macrophages have distinct niches in the heart; each of which has a diverse influence on the other cell types in the heart through cell-cell contact and secretion of inflammatory molecules and proteins. Resident cardiac macrophages in mice, lack the expression of Ly6c, are primarily derived from yolk-sac progenitors, and maintained independent of monocyte recruitment through local proliferation [13,14].

The primary resident macrophage populations are Ly6c^−^ and express Timd4, Lyve1, and either major histocompatibility complex (MHC)-II^hi^ or MHC-II^lo^ [14,15]. Interestingly, Epelman and authors identified a small population of the resident macrophages in mice, around 2%, found primarily in the cardiac microvasculature that retain Ly6c expression [14]. Ly6c- MHC-II^hi^ macrophages are CX3CR1^+^, CD206^+^, and a small subset that is CD11c^hi^, suggesting additional heterogeneity [14]. The MHC-II^lo^ population is also CX3CR1^+^ except at lower levels and are CD206^hi^ and CD11c^lo^. Dick et al. used single cell transcriptomics to demonstrate the self-renewing capacity of TIMD4^+^LYVE1^+^MHC-II^lo^CCR2^−^ cells and their role in the regulation of homeostasis and functions including endocytosis, lysosome function, and angiogenesis [16]. Similarly, Epelman et al. found that MHC-II^hi^ and MHC-II^lo^ CD11c^lo^ resident macrophage populations largely persist independent of blood monocyte input supporting the concept that resident populations cannot be replaced after injury [14]. A study evaluating non-failing human heart biopsies by single cell sequencing showed that cardiac resident macrophages expressed *Cd163*, *Colec12*, *Mrc1* (CD206), *March1*, and *Nramp1* [17,18]. These receptors, including Nramp1, are found to guard against infection, therefore guarding the heart from pathogens while downregulating inflammation [19,20,21,22,23]. Subclustering showed two populations expressed more anti-inflammatory characteristics and receptors *Rbpj*, *F13a1*, and *Col23a1* [17]. Sex differences were also observed between male and female donors in the expression of *Ps4y1* and *Linc00278* [17].

The primary role of macrophages is thought to be regulating inflammation after infection or injury however, Swirski et al. suggested cardiac macrophages may help maintain the primary functions of the heart [5]. Due to the large energy and mechanical load of the heart, resident macrophages facilitate cell and matrix turnover, removal of cell waste, and adaptations to changes in cardiac tissue strain all of which are likely to influence the steady-state cellular network [5]. The heart has unique, organ-specific needs such as electrical conductivity and energy consumption to maintain effective contractility and relaxation [5]. Macrophages have been found to facilitate nodal conduction via connexin 43-containing gap junctions and are linked to cardiac arrhythmias in diabetic mice through secretion of IL1β [24,25]. Markers highly expressed in cardiac resident subpopulation could be key in these processes. Additional studies are needed to fully understand the role that resident macrophages play in maintaining heart function.

#### 2.1.2. Macrophage Heterogeneity in Disease: Myocardial Infarction

During the first 24 h following myocardial infarction (MI), resident cardiac macrophages within the infarct region die and are replaced by recruitment of infiltrating monocyte-derived macrophages [14,26]. For example, a subset of resident cardiac macrophages, TIMD4^+^LYVE1^+^MHC-II^lo^, are partially replaced, whereas CD11b^+^CD64^+^CCR2^+^MHC-II^hi^ macrophages were found to be fully replaced by TIMD4^−^LYVE1^–^MHC-II^hi^ monocytes [16]. Due to vascular access still being intact, monocyte-derived macrophages are found initially in the infarct border zone and move to the infarct zone over time [15]. These recruited monocyte-derived macrophages have a lifespan of about 20 h and secrete chemoattractants to initiate fibrosis and promote inflammation and angiogenesis [27]. CCR2 and CX3CR1 on monocytes facilitate this recruitment to the injury site [28]. Progenitors of macrophages change when inflammation is induced, often expressing markers of inflammatory monocytes such as Ly6C^high^ in the mouse and CD14^+^CD16^−^ in humans [15].

One pathway that facilitates macrophage extravasation is through the release of prostaglandins and endothelial cell growth factor that in turn stimulates angiotensin II (Ang II) and increases vascular permeability [29]. Monocyte expression of Ang II receptor, AT_1,_ stimulates recruitment of macrophages from the spleen into the infarct [30,31]. AT_2_ is less abundant in healthy adults but is significantly upregulated in instances of cardiac injury and heart failure [32]. The role of AT_2_ however, is still unclear though it is believed to play a beneficial role by antagonizing the effects of AT_1_ [33]. As chronic inflammatory responses have been shown to lead to a worse prognosis in MI and cases of heart failure, development of antagonists of Ang II has been a primary area of focus.

In a mouse model of permanent occlusion, cardiac macrophages have distinct gene expression profiles during the first week post-MI with metabolic reprogramming being critical for polarization [34]. Multiple studies found that after MI, CCR2^+^ monocyte-derived macrophages are made up of subsets containing genes involving hypoxia, extracellular matrix (ECM) interactions, and glycolytic metabolism [16,35,36]. Early post-MI (i.e., Day 1), macrophages express a unique pro-inflammatory, ECM-degrading signature including genes associated with interleukin (IL)-1, tumor necrosis factor (*Tnf*), nuclear Factor kappa-light-chain-enhancer of activated B cells (*Nfκb*), mitogen-activated protein kinase (*Mapk*), signal transducer and activator of transcription (*Stat*)*5*, and suppressor of cytokine signaling (*Socs*)*2* pathways [27]. During post-MI days 3–7, macrophages play a key role in removal of cell debris through phagocytosis [27]. Macrophages express receptors that mediate phagocytosis early post-MI, with CD36 being expressed before MerTK [37]. Matrix metalloproteinase (MMP)-9 can cleave CD36 resulting in an attenuation of macrophage-mediated removal of necrotic and apoptotic cells [38]. Without removal of the necrotic cardiac myocytes, inflammation may be prolonged leading to adverse cardiac would healing. In male mice, genes *Folr2*, *Cbr2*, and *Mrc1* were upregulated indicating an anti-inflammatory environment at post-MI day 7 [39]. During the resolution phase at post-MI day 11, Dick et al. identified 7 unique post-infarct macrophage clusters that possessed a high degree of plasticity [16]. Despite this high degree of plasticity, none of the subpopulations adopted the core transcriptional signature of resident cardiac macrophage populations.

Macrophages also coordinate with fibroblasts to stimulate ECM production in order to stabilize the LV and prevent scar rupture [15]. Macrophages secrete high levels of transforming growth factor-β1, which drives transcription of alpha smooth muscle actin in cardiac fibroblasts [40,41]. In addition, macrophages not only serve as indirect mediators of ECM remodeling, but as a direct source of ECM components [34,42,43]. Whether or not this subset of macrophages that produce ECM are beneficial or not is unknown. Further studies are needed to determine how they influence the cardiac scar.

Macrophages are critical in the heart during homeostasis for housekeeping but can be detrimental during cardiac wound healing depending on the phenotype developed from infiltrating monocytes. While targeting macrophages may seem like a reasonable therapeutic target, Dick et al. found that depletion of resident macrophages in mice worsened their prognosis with no improvement in cardiac wound healing [16]. This indicates that therapies need to target specific macrophage phenotypes and likely, specific time points post-MI.

#### 2.1.3. Macrophage Heterogeneity in Disease: Pressure Overload

An appreciation of a causal role of inflammatory cells in the development of cardiac fibrosis in response to PO is emerging with the role of macrophages being the most well-studied. In human tissue, Hulsmans et al. reported an increase in myocardial macrophages in cardiac biopsies from patients diagnosed with HFpEF in comparison to referent controls [44]. Unfortunately, a current limitation in understanding the role and heterogeneity of immune cells in disease progression associated with increased left ventricle PO is the lack of available serial biopsies from patients. Establishing a time course of immune cell heterogeneity profiles that correlates with progressive fibrosis is critical and thus the use of animal models of PO, although imperfect, afford insight into the role of immune cells in PO-induced fibrosis of the heart.

Although contradictory, there are data from animal models to support a role of both classical, pro-inflammatory and nonclassical, anti-inflammatory macrophages in activating cardiac fibroblasts in response to PO. This may be because multiple pathways lead to activation of fibroblasts and different macrophage populations might use distinct pathways especially in the context of the method used to induce PO, i.e., Ang II versus transverse aortic constriction (TAC). For example, using TAC, a murine model of PO, expression of microRNA-21 (mir-21) in macrophages was shown to induce a pro-inflammatory macrophage phenotype [45]. Furthermore, mice with deleted macrophage expression of miR-21 demonstrated reduced fibrosis and fewer myofibroblasts suggesting that pro-inflammatory macrophages were primary drivers of fibrosis [45]. In another model of PO using Ang II infusion, Falkenham et al. demonstrated that depletion of macrophages using clodronate liposomes reduced levels of αSMA+ myofibroblasts, decreased collagen content, and decreased levels of mRNA encoding *Cxcl3*, a nonclassical macrophage marker supporting an essential role of non-classical macrophages in fibrosis. In fact, Ang II-induced PO in *Cx3cr1^−/−^* mice showed a deficiency in recruitment of non-classical monocytes which resulted in an exacerbation of cardiac fibrosis accompanied by increases in classical, pro-inflammatory macrophages, a result seemingly contradictory to that of the miR-21 studies [46]. Hence, current results seem contrary as to the primary phenotype of cardiac macrophages that drive the activation of resident fibroblasts to promote collagen deposition in PO. However, importantly, similar to post-MI, the plasticity of macrophage populations is likely fluid in nature. Cardiac macrophage populations might be better represented as multiple intermediate phenotypes that demonstrate markers for both classical and nonclassical macrophages.

Efforts to parse out contributions from resident versus recruited macrophages in PO have also been carried out. As an example, antibody-based depletion of CCR2^+^ monocyte-derived macrophages, a recruited population in early PO, 3 to 5 days following TAC, was shown to protect the myocardium from pathological remodeling and progression to interstitial fibrosis demonstrating that recruited macrophages, CCR2^+^ and Ly6C^high^, exacerbate the remodeling response to PO [47]. Macrophages likely also play an important role in producing factors that elicit recruitment of other inflammatory cell types to the myocardium. In response to TAC, myocardial levels of cytokines including CCL12, CXCL9, CXCL10, were increased, along with intracellular adhesion molecule (ICAM)1 expression on myocardial vasculature [48]. Ngwenyama et al. demonstrated that recruited and resident macrophages are the primary producers of CXCL10 and CXCL9 for the recruitment of CD4^+^ Th1 cells which exacerbate pathological remodeling of the myocardium, and Patel et al. demonstrated that antibody-based depletion of CCR2^+^ monocyte-derived macrophages decreased CD4^+^ T-cell populations at 4-weeks post-TAC [47,49]. A potential therapeutic direction for PO could be to target recruited macrophages to reduce the secretion of cytokines, such as CXCL9 and CXCL10, thus limiting recruitment of CD4^+^ Th1 cells and perhaps slowing the progression of heart failure. Further characterization of myocardial macrophage populations through single cell analysis, flow cytometry, and imaging mass cytometry in a time dependent manner will facilitate more precise identification of macrophages and their role in fibrosis.

#### 2.1.4. Macrophage Heterogeneity: Summary

Macrophages are critical in both health and disease in the heart. Yolk-sac derived cardiac macrophages regulate endocytosis, lysosome function, angiogenesis, and regeneration. Post-MI, the majority of yolk-sac derived cardiac resident macrophages within the ischemic zone are replaced by monocyte-derived macrophages. These populations promote inflammation and fibrosis in the infarcted area, facilitating in development of heart failure. In contrast, following PO, both resident and recruited monocyte-derived macrophages play a key role in interstitial fibrotic development and activation of other cell types including fibroblasts and CD4^+^ T-cells. Designing a more targeted therapy that inhibits the detrimental effects without impeding the beneficial roles of macrophages could improve both cardiovascular pathologies.

### 2.2. Dendritic Cells

#### 2.2.1. Dendritic Cells at Homeostasis

Dendritic cells make up about 1% of the total cardiac leukocyte population and bridge the innate and adaptive immune system [14,50]. According to their hematopoietic origin, dendritic cells are classified into two main populations: plasmacytoid (pDCs) or conventional/myeloid dendritic cells (mDCs) [51]. In the heart, dendritic cells are mostly concentrated in the aortic valve and express high levels of CD11c [14,50]. In steady state, cardiac dendritic cells function to preserve peripheral tolerance to the heart [52]. Austyn et al. found that in healthy mouse hearts, dendritic cells resemble a more immature than mature population due to little stimulatory activity immediately after isolation [53]. These immature dendritic cells patrol the heart, similar to cardiac resident macrophages, and become activated when encountering foreign or pathogenic antigens or markers of tissue damage and inflammation [54]. Phagocytosis of antigens by immature dendritic cells results in an increase in the maturation marker CD83 and Class I and Class II MHCs [55].

#### 2.2.2. Dendritic Cell Heterogeneity in Disease: Myocardial Infarction

More recently, there has been a greater interest in the role of dendritic cells after MI. Fukui et al. demonstrated that patients admitted to the hospital after acute MI had fewer numbers of circulating mDCs and pDCs compared to patients with stable angina [56]. These numbers however, returned to levels comparable to control 7 days after the initial event and remained stable for the next 3 months [56]. This same trend of lower DCs in circulation immediately after hospitalization was found in multiple studies [56,57,58]. Interestingly, Wen at al. found that the percent of circulating mDC precursors negatively correlated with severity and extent of coronary artery lesions [58]. Immunohistochemistry of post-mortem tissue from patients with MI indicated that the reduction of circulating dendritic cells was likely due to recruitment into the infarcted myocardium [56]. Furthermore, the number of CD209^+^ and CD11c^+^ dendritic cells within the infarcted myocardium correlated with the extent of reparative fibrosis suggesting this influx may be beneficial for the healing response [59]. In a mouse model of MI, all dendritic cell subsets infiltrated the heart, except for a subset of mDCs, which migrated to the mediastinal lymph node and presented cardiac self-antigen to autoreactive CD4+ T-cells, which adopted a Th1/Th17 effector phenotype [52]. Post-infarct autoimmunity may stimulate persistent myocardial inflammation, leading to further damage with long-term pathological consequences.

#### 2.2.3. Dendritic Cell Heterogeneity in Disease: Pressure Overload

The role of dendritic cells in myocardial remodeling during PO is not well defined. Martini and Kunderfranco demonstrated that following 1 or 4 weeks of TAC, the myocardial dendritic cell population, expressing CD209 and MHCII, was decreased [60]. However, Laroumaine et al. demonstrated that the chemokine *Ccl17*, expressed mainly by dendritic cells, was upregulated in the myocardium following TAC [61]. Further research is necessary to understand the role of dendritic cells in response to PO. The use of Mgat2, a dendritic cell specific knockout mouse, subjected to TAC, might be a useful tool to elucidate the role of dendritic cells in exacerbating or attenuating the response to PO.

#### 2.2.4. Dendritic Cell: Summary

In a healthy heart, dendritic cells are found mostly in the aortic valve and facilitate the development of peripheral tolerance. Post-MI, the extent of dendritic cell function is unknown although studies have demonstrated increased activation of dendritic cells in the infarcted myocardium. The literature regarding the role of dendritic cells in response to PO is conflicting with some studies showing fewer dendritic cells after TAC and others showing elevations. Additional studies are needed to define the role of these cells in promoting the progression of disease.

### 2.3. Neutrophils

#### 2.3.1. Neutrophils at Homeostasis

There is a knowledge gap with regards to neutrophil cell heterogeneity in the heart during homeostasis. While most of what we know on the role of neutrophils is within the setting of disease, there is some evidence to suggest that neutrophils may also be present in low abundance in a healthy heart. Martini et al. showed using single cell sequencing that there is a small proportion of neutrophils present in the healthy heart [60]. Through detection of CD39 and CD73, which are highly expressed on neutrophils, Bönner et al. showed that there were about 2.3 × 10^3^ resident neutrophils/mg tissue in a control mouse heart [62]. Interestingly, although neutrophils from steady state hearts all mapped to post-MI subgroups, they did not show enrichment for sialic acid binding Ig-like lectin F (*Siglecf*), *Icam1*, or *Tnf* [35]. Whether these neutrophils are truly resident cells or just patrolling cells that may have been caught in the vasculature at the time of tissue collection is unclear. Better assessment of the tissue is necessary to determine the root of these cells found in the heart at steady state.

#### 2.3.2. Neutrophil Heterogeneity in Disease: Myocardial Infarction

After a MI, neutrophils infiltrate the heart, peaking 24–36 h after the ischemic event [15]. Molecules released by neutrophils such as neutrophil gelatinase associated lipocalin (NGAL or LCN2) have been linked to adverse outcomes in chronic heart failure. Plasma NGAL levels are correlated with MMP-9 in post-MI patients [63]. NGAL protects MMP-9 from degradation resulting in increased MMP-9 activity and ECM degradation [64]. Myeloperoxidase (MPO) is a protein secreted by neutrophils that has potential to be diagnostic in patients with acute MI [65]. Ali et al. found that inhibition of MPO in a MI mouse model led to decreased dilation of the LV after 7 days of treatment and improved cardiac function and remodeling after 21 days of treatment likely due to an attenuation of recruited inflammatory cells [66]. Tracchi et al. illustrated that patients with chronic heart failure had an increased neutrophil lifespan in plasma [67]. Interestingly, anti-neutrophil therapies have been shown to worsen cardiac function, increase fibrosis, and increase risk for heart failure [68]. This highlights that while excessive neutrophils can be detrimental, they are needed for beneficial healing of the infarct.

Through time-series single cell genomics and cellular indexing of transcriptomes and epitopes by sequencing (CITE-seq), Vafadarnejad et al. revealed different subsets of neutrophils after MI [35]. At Day 1, neutrophils were characterized to be derived from bone marrow neutrophils with putative activity involved in hypoxic response and emergency granulopoiesis [35]. At Days 3 through 5, 2 major subsets of neutrophils were identified, SIGLECF^hi^ and SIGLECF^low^. It is believed that the SIGLECF^low^ group represents young blood neutrophils, whereas SIGLECF^hi^ neutrophils are acquired within the ischemic heart tissue and represents tissue differentiated neutrophils [35,69]. Calcagno et al. found SIGLECF^hi^ neutrophils are Myc-recovered, NFκB-activated, involved in survival signaling, and resistant to apoptosis suggesting that this subpopulation of neutrophils are a comparatively long-lived population within the infarct [69].

Through proteomic analysis of isolated neutrophils from mouse infarcts, Daseke et al. also demonstrated that neutrophils have temporally dependent roles [70]. Day 1 neutrophils have increased MMP activity, while Days 3 and 5 neutrophils have induction of ECM organization [70]. Furthermore, Day 7 neutrophils induce scar formation through increased expression of fibronectin, galectin-3, and fibrinogen [70]. In a similar study, neutrophils were shown to express markers similar to what is observed on macrophages with the more proinflammatory (N1) markers at Day 1 post-MI and anti-inflammatory (N2) markers at Day 7 [71,72]. Although the majority of neutrophils detected in the infarct region were CD206^−^, a relative increase of the CD206^+^ N2 neutrophil subset was found in the infarcted tissue [71]. CD206^+^ N2 cells were absent from the circulation indicating this phenotype is activated by the infarct microenvironment [71]. The role of these Ly6G^+^CD206^+^ neutrophils post-MI is not clear however, based on studies in cancer and stroke, it is likely that these neutrophils exhibit pro-angiogenic or pro-fibrotic actions [73,74].

#### 2.3.3. Neutrophil Heterogeneity during Disease: Pressure Overload

Single cell sequencing, after 4 weeks of TAC in mice, identified two clusters of neutrophils present in the myocardium, recruited in response to PO [60]. Similar to observations in post-MI hearts, one population of neutrophils expressed proinflammatory genes, whereas the second population demonstrated an anti-inflammatory, pro-repair phenotype characterized by the expression of *Mmp9* and *Arg2* genes [60]. Although at lower numbers compared to MI, Calcagno et al. identified both SIGLECF^hi^ and SIGLECF^low^ neutrophil populations [69]. When neutrophils were depleted from the myocardium using a Ly6G antibody, total tissue mRNA levels of *Il6* and *Il1β* were reduced along with a decrease in infiltrating monocytes and macrophages demonstrating a plausible role for neutrophils in recruiting specific immune cell types to the myocardium [75]. Furthermore, decreased hypertrophy and posterior wall thickness, and improved fractional shortening were demonstrated with Ly6G antibody neutrophil depletion, although the impact on fibrosis was not determined [75]. Targeting neutrophils in PO might be beneficial as this cell type could orchestrate the recruitment of macrophages, which in turn, are implicated in recruitment of T-cells to the myocardium.

#### 2.3.4. Neutrophil Heterogeneity: Summary

In summary, while there is limited literature on neutrophil function in the healthy heart, there is evidence that they are present in low abundance. Post-MI, neutrophils are primarily pro-inflammatory though some evidence suggest that their function alters slightly during the remodeling time course. Anti-neutrophil therapies have been found to negatively impact cardiac function, but inhibition of proteins secreted by neutrophils have shown some promise. Following PO, neutrophils are recruited to the myocardium and exacerbate the hypertrophic remodeling of the heart. In contrast to post-MI studies, antibody-mediated depletion of neutrophils after PO has shown promise in recovering PO-mediated cardiac dysfunction. The role neutrophils play in interstitial fibrotic progression remains to be elucidated.

### 2.4. T-Cells

#### 2.4.1. T-Cells at Homeostasis

T-cells are part of the adaptive immune system classified by the expression of CD3 and become activated by antigen presenting cells including dendritic cells and macrophages [76]. T-cells can be further divided into subsets based on additional receptor markers and physiological processes [76]. CD4+ T-cells can be divided into helper T-cells (Th1, Th2, Th17), and regulatory T-cells (Tregs). CD8+ T-cells are cytotoxic T-cells and are mainly known for killing pathogens, infected cells, and cancerous tumors [76]. Tucker et al. discovered T-cells present in the healthy human heart expressed *Cd2, Cd69*, and *Trat1* [17]. Casey et al. identified an effector resident memory CD8+ T-cell population expressing CD69 and CD103 in the human and mouse heart, although their role is not yet known [77]. It can be hypothesized that these CD8+ T-cells play a role in conductivity as CD69 has a known role in calcium ion binding [78].

In order to prevent T-cell activation and recruitment during homeostasis, T-cells undergo positive selection for MHC receptors with a low affinity for self-peptides [79]. This selection process is called self-tolerance. Another preventative measure is the requirement of three separate signals for full T-cell activation. Antigen-loaded antigen-presenting cells bind the target T-cell to initiate cell activation. CD4+ T-cells then bind to CD80/CD86 via CD28 while CD8+ T-cells require CD70/CD137 to stimulate clonal expansion [80]. Lastly, cytokines provide more detailed instructions for the T-cells to perform specific tasks such as direct killing of the pathogen or regulating inflammation [80]. These processes ensure that T-cells do not become autoreactive and alter cardiac homeostasis.

#### 2.4.2. T-Cell Heterogeneity in Disease: Myocardial Infarction

Disease transforms cardiac homeostasis via loss of self-tolerance and stimulation of T-cell exhaustion. With a loss of self-tolerance, T-cells recognize self-antigens as foreign and become activated and attack self, causing an increase in inflammation and autoimmunity [81]. T-cell exhaustion is the progressive loss of T-cell function and depletion of T-cell numbers over time with chronic infection or inflammation [82]. Prolonged or mistimed inflammation after MI can lead to either T-cell exhaustion or a decrease in self-tolerance, leading to an imbalance in the immune system and development of disease [52,83].

Damage associated molecular patterns (DAMPs), chemokines, and other inflammatory mediators activate antigen presenting cells which then present the antigens to T-cells. This stimulation causes the T-cells to leave the lymphoid organs (e.g., thymus, spleen, lymph nodes) and traffic to the heart through blood and lymphatic vessels [84]. In mouse and rat models, CD4+ T-cells infiltrate the heart during the first week post-MI and contribute to myocyte injury through release of IFNγ [85,86]. At day 7 post-MI, more than 50% of the total T-cell population were CD8+ T-cells followed by Th1 and Tregs as the second largest population [87]. By 8 weeks post-MI, T-cell phenotypes are more evenly distributed with T-regs and Th2 both making up 25% each of the infarct [87]. Youn et al. found that in male patients with acute heart failure, there was an increase in CD4 + CD57+ T-cells compared to healthy patients [88]. CD4 + CD57+ T-cells were shown to be associated with increased inflammatory markers such as IFNγ and TNFα, leading to prolonged and possibly mistimed inflammation and adverse cardiac wound healing [88].

While Tregs are not found in the healthy myocardium, Treg numbers peak at Day 7 after MI and have been shown to be protective [89,90,91,92,93]. Dobaczewski et al. found deletion of CC chemokine receptor 5 (Ccr5) resulted in a flawed inflammatory response to the infarct region in a mouse model of reperfused MI whereby Tregs had decreased infiltration to the infarct [94]. Additionally, Ccr5 deficient animals displayed prolonged inflammation that led to adverse cardiac remodeling [94]. Xia et al. illustrated Tregs that accumulate in the injured mouse myocardium after MI have a transcriptome distinct from lymphoid organs [90]. These novel features included a group of ECM organization or collagen synthesis-related genes including, secreted acidic cysteine-rich glycoprotein (*Sparc*), periostin (Postn), collagen (*Colla1* and *Col3a1*), and fibronectin (*Fn1*) which were strikingly upregulated [90].

In murine models, gamma delta T-cells are found within the infarct as early as Day 1 post-MI and peak at Day 7 post-MI [95]. Funken et al. found that the loss of gamma delta T-cells in mice helped improve organ injury [96]. Through a gamma delta T-cell genetic depletion mouse model, Caillon et al. discovered that this T-cell subset may also mediate vascular injury and activation of other T-cell subsets thus fostering an exacerbation of inflammation [97].

In contrast to Tregs, CD8+ T-cells have been found to be detrimental post-MI [85,98]. Plasma levels of granzyme B, a CD8+ T-cell secreted protein, correlated with left ventricular end-diastolic diameter in post-MI patients [99]. Santos-Zas et al. demonstrated that CD8+ T-cells are recruited to the infarct and numbers peaked at day 3 post-MI [85]. This conflicts Yan et al. who demonstrated CD8+ T-cells were present 1 day after MI, although at low numbers, and peaked around Day 7 post-MI [100]. Bansal et al. found in mice that CD8+ T-cell counts, in addition to CD4+ T-cells, remain elevated 8 weeks post-MI [101]. Using CD8+ T-cells that lacked the ability to secrete granzyme B, Santos-Zas et al. demonstrated CD8+ T-cells stimulated adverse remodeling and decreased cardiac function at day 21 post-MI by granzyme B mediated mechanisms [85]. Similarly, Ilatovskaya et al. found using a genetic model of CD8+ T-cell depletion, mice lacking CD8+ T-cells had improved cardiac function and survival rates 7 days post-MI compared to WT mice [98]. In this study, CD8+ T-cells were found to play a role in activating macrophages to facilitate removal of necrotic debris [98]. In contrast, mice that received 28 days of lipopolysaccharide before MI had an upregulation of CD27+ expression on CD8+ T-cells in the infarct compared to MI controls suggesting activation of the memory response [102]. This increase in memory, however, did not affect macrophage recruitment or activation at Day 1 post-MI.

CD8 + CD57+ T-cells, a subset involved in replicative senescence, correlates with cardiovascular mortality 6 months after acute MI in patients [103]. CD57+ T-cells are highly vulnerable to activation-induced apoptosis and fail to proliferate after in vitro antigen-specific stimulation [104]. Due to their pro-inflammatory and high cytotoxic capacities, CD8 + CD57+ T-cells most likely regulate acute coronary events. In contrast, CD8+ T-cells that express the angiotensin type 2 receptor (AT_2_R) have been shown to have no cytotoxic activity and are believed to have a cardioprotective effect after MI [104]. These studies highlight the complexity of T-cell biology in post-MI remodeling and that a balance is needed between the T-cell phenotypes to reduce harmful inflammation.

#### 2.4.3. T-Cell Heterogeneity in Disease: Pressure Overload

RAG2^−/−^ mice that lack functional B- and T-cells do not progress to heart failure following TAC surgery, supporting a critical role of lymphocytes in mediating disease severity [105]. CXCR3 is primarily a marker of CD4+ Th1 cells and recruitment of CXCR3^+^ CD4^+^ Th1 cells, in response to TAC, was shown to lead to exacerbated pathological remodeling and interstitial fibrosis. CXCR3^−/−^ mice exhibited reduced interstitial fibrosis, reduced cardiomyocyte hypertrophy, and preserved cardiac function [49]. In a separate study, stimulated mediastinal draining lymph node cells isolated from 6-week TAC mice had decreased expression of IL-4, whereas IFNγ was increased, representative of a Th1 polarization of T-cells in TAC mice [61].

Furthermore, T-cells are implicated in playing an important role in promoting myocardial fibrosis through the regulation of lysyl oxidase (LOX), a prominent collagen crosslinking enzyme. Notably, RAG2^−/−^ mice were reported to have decreased expression of mature LOX [61]. When T-cells were replenished (RAG2^−/−^ with CD3+ cells), levels of both proLOX and LOX protein were significantly increased [61]. Other crosslinks, such as those generated by transglutaminase and by advanced glycation end products (AGEs), play an important role in the fibrotic myocardium, but it is not currently known if T-cells are a primary effector of these crosslinks in response to PO.

In a recent study, Komai et al. identified a previously unknown role for CD8+ T-cells in regulation of cardiac-resident macrophages and infiltrating macrophages [106]. The authors determined that CD8+ T-cells converted both macrophage subsets into cardioprotective macrophages secreting growth factor genes such as amphiregulin, oncostatin M, and insulin-like growth factor-1, which have been shown to be essential for the myocardial adaptive response after PO. A paucity of current therapies to treat fibrosis are currently available. One potential therapeutic opportunity to be considered is targeting T-cells as mediators of cardiac fibrosis in PO.

#### 2.4.4. T-Cell Heterogeneity: Summary

There is a limited number of T-cells present in the healthy heart however, their role is not quite understood. While the stimulus for T-cell activation is likely different between MI and PO, there are quite a few similarities in the role T-cells play during these pathological processes. Post-MI, different T-cell subsets are present in a temporal dependent manner. Each T-cell subset has a distinct role in regulating the post-MI remodeling process ranging from activation of innate immune cells to having a direct impact on cardiac fibrosis. In response to PO, T-cells are recruited to the myocardium and exacerbate the development of interstitial fibrosis by secreting crosslinking enzymes that modify collagen to make it more insoluble and resistant to degradation.

### 2.5. Understudied Immune Cells in MI and PO

While the homeostatic and pathogenic role of B-cells and natural killer (NK) cells are less understood than other cell types, there is evidence of the presence of these cells in the heart during homeostasis and in pathological conditions. Martini et al. found that B-cell subpopulations were abundant in both healthy and PO murine hearts [60]. In PO mice, NK cell numbers expanded slightly compared to the healthy hearts and had a significant association with immune activation and cytokine production pathways [60]. Patients with MI have been shown to have a slight decrease in the number of circulating B220+ B-cells within an hour after myocardial reperfusion, which was followed by a significant increase 24 h after reperfusion [107]. According to Zougarri et al., B-cells post-MI produce Ccl7 to induce Ly6C^hi^ monocyte recruitment to the infarcted heart and impaired myocardial function [108]. Literature describing NK cells post-MI is currently conflicting. While one group has found NK cells to promote vascularization and angiogenesis post-MI, these findings have not been repeated [109]. Additional research is needed to fully understand the role of B-cells and NK cells in the healthy and diseased state of the heart.

## 3. Technologies Used to Discover Immune Cell Heterogeneity

Studying immune cell heterogeneity in the heart before and after injury is quite complex. Many immune cells express the same cell surface receptors and proteins making it tough to distinguish certain immune cell groups from others. For example, CD44 is expressed on both T-cells and macrophages. CD44 has many roles in proliferation, migration, and lymphocyte activation [110]. Similarly, CD8, which is a key marker for cytotoxic T-cells, is also expressed by dendritic cells [111]. Because of the overlap in immune cell markers to classify cells into the correct subgroups, multiple markers should be used to positively capture the desired target cell population. Recent advances in technology used to study cell heterogeneity have made it easier to isolate and characterize immune cell populations with increased precision and accuracy. Below we discuss technological advancements for understanding cellular heterogeneity in models of cardiac disease and summarize the advantages and limitations of these techniques (Table 1). Understanding the limitations of each method and how different techniques complement one another is vital when designing experiments to study the role of the immune system in the heart. Nonetheless, great promise is seen in utilizing new techniques for the future of cardiac research and eventually, the development of novel therapies.

### 3.1. Flow Cytometry

Flow cytometry is used to detect and measure physical and chemical properties of isolated cells. Utilizing antibodies for specific immune cell classification and phenotypes can define the immune cell composition within tissue, serum, plasma, etc. Flow cytometry can also be used to phenotype cells by measuring cellular proliferation, and viability, in addition to intracellular cytokine and signaling proteins, and cell cycle stages [112].

Fluorescent activated cell sorting (FACS) is a way to enrich for a cell population of interest from a sample by using flow cytometry technology. The cells of a pre-selected population can be isolated and sorted rapidly with high purity; some reports indicate a purity of up to 99% [112,113]. Sorting is performed by giving charges to droplets containing single cells. Single cells are then detected by an electric field and sorted into collection tubes according to their charge [112]. After sorting, enriched cells are often sent for sequencing, biochemical analysis, or grown in culture.

#### Strengths and Limitations

Flow cytometry and FACS techniques are fairly easy to use, highly translational, and affordable [112]. Because flow cytometry data removes debris and dead cells, the accuracy is much better than more traditional techniques such as immunofluorescence. One limitation of flow cytometry and FACS is that these techniques are dependent on antibodies or stains to characterize the cells. In addition, because the number of detectors/filters available to use are limited, some immune cell populations will be difficult to analyze. However, recent advances have greatly increased the number of fluorescently tagged antibodies available for flow cytometry. Available fluorophores with distinct excitation and emission spectra along with multicolor flow cytometers have enhanced the ability to generate multidimensional expression data but choosing fluorochromes with minimal spectral overlap to fully characterize the cell sub-populations remains tricky [112]. Online resources such as Fluorofinder, Spectrum Viewer, Fluorescence Spectra Viewer, or the Spectra Analyzer can help design antibody panels and assess the degree of spectral overlap and potential spillover. Given the lack of reagents for most animal models, these advanced techniques have mainly benefitted the analysis of human and mouse cell samples. In addition, because samples are in suspension, spatial and cellular interaction data are not easily attainable. Overall, flow cytometry and FACS techniques have contributed considerably to identifying cellular phenotypes of immune cell populations in the heart.

### 3.2. Mass Spectrometry

Mass spectrometry and its applications are being used more readily at the bench and clinic. Matrix-assisted laser desorption/ionization- mass spectrometry (MALDI-MS) is an extremely sensitive technique used for high throughput proteomic assessment of tissue and cellular samples which uses the intensity of the spectrum to determine the abundance of the compound [114,115]. MALDI imaging mass spectrometry (MALDI-IMS) is a specialized proteomic method that can identify the molecular composition and abundance while also giving spatial distribution of proteins in a tissue section [116]. MALDI-IMS measures the mass spectra at specific spatial points on the sample to provide a hyperspectral image with a corresponding mass spectrum at each measured pixel, which then corresponds to specific compounds and their location in the sample [115]. MALDI-IMS is useful to visualize changes in the microenvironment including alterations in chemokines, ECM, and growth factors, that might directly influence cellular recruitment and activation, leading to impaired states of wound healing.

Cytometry by time of flight (CyTOF), is a powerful single-cell proteomic analysis technique which utilizes rare metal isotopes instead of fluorophores as antibody tags. The application of CyTOF is similar to flow cytometry and often used to quantify labeled intracellular proteins or cell surface targets. Because CyTOF uses metal isotopes instead of fluorophores, a larger number of markers can be assessed while remaining high-dimensional and unbiased [117].

#### Strengths and Limitations

Mass spectrometry detection is flexible as it can detect proteins, lipids, cell metabolism, and drug metabolites on one platform making this method applicable to many fields [118]. The advantages of using proteomic techniques are high speed data acquisition, off-line analysis capabilities that can connect to liquid chromatography (LC) separation, easy sample preparation with controllable sample consumption, and the ability to archive and recover samples [114,118,119,120,121]. Gadalla et al. compared flow cytometry with CyTOF and found that CyTOF was highly accurate, even in assessing low immune cell numbers [117]. From this investigation, they were able to develop a 40+ parameter panel for CyTOF which was used for broad scale immune cell profiling and biomarker discovery [117].

Proteomic techniques, however, are expensive, and access to a core with a mass spectrometer capable of running and analyzing proteomic data are not always available. Due to ion detection mechanisms, saturation effects, and signal to noise limitations, varied results are sometimes generated between instrumentation despite similar sample preparation protocols and instrument conditions [122]. Instrument variability can sometimes lead to proteins being masked in one system but be within the discoverable range in others.

### 3.3. Single Cell Sequencing

Immunology has greatly benefited from single cell sequencing which uses genomic techniques to evaluate the sequence of individual cells rather than the average of whole cell populations. Single cell sequencing evaluates the heterogeneity of cell populations to determine differences in genetics, maturity, or antigen presentation in addition to the host response, in turn giving more insight into how the immune system acts on an individual level within the host [123]. Because not all cells of a population have the same genetic sequence due to somatic mutations and methylation/epigenetic patterns introduced during DNA replication, single cell sequencing could lead to a boom of personalized medicine as it becomes more widely available and affordable [124].

#### Strengths and Limitations

One strength of single cell sequencing is the ability to perform multiomics analysis. Genomic, epigenetic, and transcriptomic data from the same cell set is more reliable than overlaying single data sets from multiple experiments due to fewer batch effects and decreased sampling bias [125]. One of the major challenges for single cell sequencing is that the initial isolation likely results in changes to cellular physiology. Cells are heavily influenced by their surrounding environment and cellular interactions. Single cells in suspension are no longer in their native environment, which could result in unintended cellular stress, altering the behavior, viability, and genetic profile of the cell [126]. Cell viability is required for downstream analysis, and new technologies are becoming available that allow for the isolation of cells in a gentle manner to ensure viability. These include cell printers and microfluidic platforms that harness picodroplet technology, which protect individual cells from shear stress.

Another caveat with single cell sequencing is the analysis of these data sets can be challenging and likely need a bioinformatician due to the nature of the multidimensional data generated [125]. In addition, single cell sequencing is more expensive than traditional bulk RNA sequencing. While single cell sequencing data on its own cannot provide spatial information, there are computational techniques that can be used to map out spatial data. Cellular clusters in single cell RNA-seq experiments, when coupled with immunohistochemistry staining, can provide important spatial information regarding the location of these cell types in the heart [125,127,128,129]. In addition, recent advancements have made it possible to determine spatial transcriptomics with as small as a 2 μm resolution by utilizing novel spatial barcoding technology on histological tissue sections for library preparation [130]. How well this new technology can capture some of the more minor populations such as T-cell subsets in the post-MI heart has not yet been studied.

## 4. Conclusions

Cells in the heart, both in health and disease are constantly being kept in check through the innate and adaptive immune system. Improving our understanding of cell heterogeneity in a healthy and diseased heart will facilitate a more targeted treatment for detrimental cell populations without effecting the activity of beneficial cells. Despite great advancements the field has made, there is still more to learn. Conflicting evidence of immune cell populations and their abundance and role in maintaining heart homeostasis needs to be clarified to better understand the role of different immune cell populations in disease. Some immune cell subsets also lack a common naming system, leading to lack of clarity when determining which markers to use to determine cell groups of interest. In addition, there is a need to better understand the role of the cell populations during homeostasis and how these immune cell populations interact and influence each other. Due to differences in pathology, current research in PO animal models is focused on reversing fibrosis, while research concerning MI aims at determining what components (e.g., amount and type of extracellular matrix) make up a beneficial scar [131]. Understanding the differences and similarities between the immune cell profiles of these two disease states will facilitate the development of potential immunotherapies.

## Figures and Tables

**Figure 1 biomolecules-12-00011-f001:**
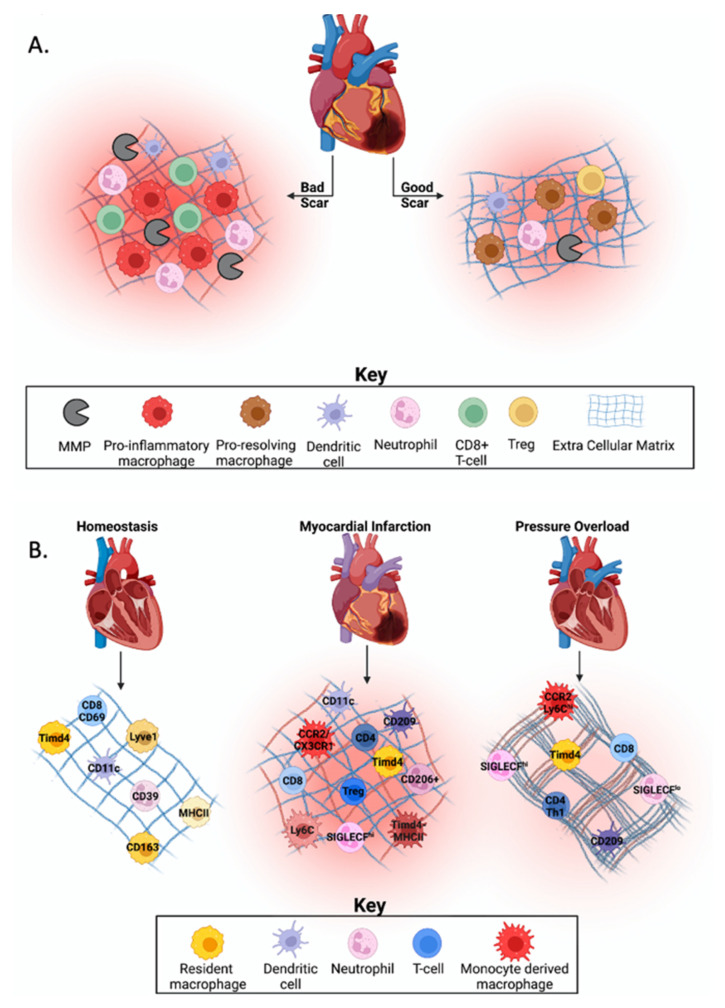
Immune cell heterogeneity in cardiovascular homeostasis and disease. (**A**) The Goldilocks zone post-myocardial infarction (MI) is defined by the balance of immune cells needed for cardiac healing. Excessive or prolonged inflammation may lead to increased cardiac rupture due to disproportionate collagen deposition, infarct expansion through phagocytosis of healthy cardiomyocytes, or increased dilation of the left ventricle, which leads to an increased risk of developing heart failure. (**B**) During homeostasis, yolk-sac derived resident cardiac macrophages are the dominant immune cell type, with evidence that dendritic cells, neutrophils, and T-cells are also present, but in low abundance. Post-MI, resident cardiac macrophages are replaced by monocyte-derived macrophages, which have a pro-inflammatory phenotype. There is also an increase in dendritic cell activation (e.g., CD209) and abundance. Neutrophil abundance increases while function changes over time in the infarcted myocardium, and an increase in effector T-cell subsets is seen. In models of pressure overload (PO), resident and recruited macrophages, play a role in fibrotic development through activation of other cell types. While literature is conflicting, dendritic cells may play a role in fibrotic development. Neutrophil abundance is increased following PO, but their role in fibrotic development is unknown. After PO, T-cells are recruited to the myocardium where they secrete cross-linking enzymes that modify collagen to make it more insoluble and resistant to degradation. Created with BioRender.com.

**Table 1 biomolecules-12-00011-t001:** Advantages and disadvantages of technologies used to discover cell populations and heterogeneity in the heart and throughout the body.

Technology	Strengths	Limitations
Flow Cytometry and Sorting	Translational—used often in the clinic and wet labs Cost-effective Easy sample prep Quicker at characterizing heterogeneous cell populations	Not as powerful Cannot obtain spatial and interaction data Limited on what markers can be used to characterize cells (e.g., antibody-based) Signal spillover and contamination is possible
Mass Spectrometry	Limited sample preparation Generates data quickly Detection is flexible (i.e., proteins, lipids, metabolites) Applicable to many fields: pathology, diagnostics, pharmaceutical	Expensive Dependent on a core capable of running and analyzing data Results can vary from instrument to instrument Signal to noise limitations
Single Cell Sequencing	Characterization of an individual cell versus a whole population More reliable when performed in conjugation with other -omic techniques due to fewer batch effects and sampling	Expensive Data sets are not user friendly Spatial data requires additional techniques/expertise

## Data Availability

Not applicable.

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
