# Peer review of "Organized Chaos: Deciphering Immune Cell Heterogeneity’s Role in Inflammation in the Heart"

_biomolecules, 2021, doi:10.3390/biom12010011_

Round 1
Reviewer 1 Report
Corker et al provided a review article on the roles of several major types of immune cells in the settings of healthy and disease, myocardial infarction and pressure overload. The final section covers the current main techniques used in relevant research with critical comments included. Overall this review is well thought through and the contents are well organized. The citation of references is up-to-date with 73% of references published within the last 10 years (after 2011). I have only a few comments for authors’ consideration in revision.
Comments:
- The paper would be improved with inclusion of a couple of diagrams depicting the dynamic features of certain types of immune cells in the referred diseased setting. This addition would also increase the readability of the paper.
- It would be helpful if a brief summary note could be added to the end of each main sections on consensus as well as key questions.
- Recent studies using single cell omics have revealed sex differences and cardiac chamber-dependent differences in diverse cell types of mouse or human hearts, including immune cells. (Tucker NR, et al. Circulation 2020;142:466; Wang L, et al: Cardiovasc Res epub Apr 11; Squiers GT, et al: Cardiovasc Res 2021;117:2252). These findings would be worthwhile to be referred in this review paper.
- There are some typos (line 270 “an MI”, line 374: “distinct etc distinct”) and errors in references 44 and 47. Please check.
Reviewer 2 Report
Corker et al have set out to present a review on immune cell dynamics, and function in the healthy and diseased (mammalian) heart. This is a very valuable and interesting endeavour and quite some information and data has been gathered on the subject. However, the manuscript needs to be ordered and structured better to make it clearer and more beneficial for the reader. In it's current form I cannot endorse publication.
Some specific points:
- Abstract
- "daily house-keeping functions" --> not clear what these are!
- "... compose 5.3% of the cellular population in healthy ventricular tissue ..." --> is this review just about the ventricles? How can you define this so precisely? Which species does this refer to ... human?
- The aim of the study "... we will discuss, ..." sounds very interesting, but I unfortunately doubt this is delivered ... see below!
- Introduction
- provide a better picture of ratios (percentages) of cells in the healthy heart (which species?) vs the diseased heart (which disease?)
- if Goldilock is a relevant concept (unknown to me!), explain it better
- The specific differences of inflammation between MI, HFpEF, PO remain unclear
- Chapter 2
- The structure of addressing specific immune cell types in sub-chapters and different disease states is appealing. However, the text must be structured better, probably shortened to get the major points (and take home messages) more visible
- Make sure to distinguish speculation from evidence (e.g. line 92 "Swirski et al suggested cardiac macrophages may ...")
- Specify what you actually want to convey (e.g. line 96 "The heart has specific needs to maintain ..." --> Which needs?
- Chapter 3
- The technology outline is a good idea, but needs to be shortened and more to the point
- Table 1 delivers rather generic information
- Conclusion
- The take home messages seem rather limited
- Figures for each Chapter or even Sub-chapters (especially 2.xx) would be very helpful
Author Response
Thank you for your very helpful comments. Please see attached document.

Reviewer 3 Report
This is an interesting and extensive review on the immune cells in the heart.
I have a few minor comments. I actually thought that the first sentence was confusing. Was this to say that cardiomyocytes are less abundant than fibroblasts, macrophages, etc.? I suggest revising the first few sentences.
In the section on macrophages at homeostasis, I suggest moving the sentence that begins "Epelman and authors" (line 73) to the following paragraph. It seems out of place as it is now.
I suggest including some figures on the macrophages present in MI and pressure overload. The text is dense and some schematics would aid in understanding.
I also suggest adding a figure on T cell dynamics post MI.
Please make sure that it is clear to the reader whether the studies discussed are from mice or humans. In some cases (especially the CD8 T cell section, I found myself being confused on whether the data were from mice or humans.
Are there any studies that examine B cells, NK cells, or gamma delta T cells in MI or pressure overload. A small section that discusses the other immune cells that are less understood in the heart would be good.
Author Response
Please see attached comments.
